# Unveiling the Genetic Mechanism of Meat Color in Pigs through GWAS, Multi-Tissue, and Single-Cell Transcriptome Signatures Exploration

**DOI:** 10.3390/ijms25073682

**Published:** 2024-03-26

**Authors:** Cheng Liu, Zitao Chen, Zhe Zhang, Zhen Wang, Xiaoling Guo, Yuchun Pan, Qishan Wang

**Affiliations:** 1Department of Animal Science, College of Animal Science, Zhejiang University, 866# Yuhangtang Road, Hangzhou 310058, China; 22317081@zju.edu.cn (C.L.); barnettca@outlook.com (Z.C.); zhe_zhang@zju.edu.cn (Z.Z.); wangzhen20@zju.edu.cn (Z.W.); guoxiaoling@zju.edu.cn (X.G.); panyc@zju.edu.cn (Y.P.); 2Hainan Institute, Zhejiang University, Yongyou Industry Park, Yazhou Bay Sci-Tech City, Sanya 572000, China

**Keywords:** GWAS, meat quality, single-cell enrichment, TWAS

## Abstract

Meat color traits directly influence consumer acceptability and purchasing decisions. Nevertheless, there is a paucity of comprehensive investigation into the genetic mechanisms underlying meat color traits in pigs. Utilizing genome-wide association studies (GWAS) on five meat color traits and the detection of selection signatures in pig breeds exhibiting distinct meat color characteristics, we identified a promising candidate SNP, 6_69103754, exhibiting varying allele frequencies among pigs with different meat color characteristics. This SNP has the potential to affect the redness and chroma index values of pork. Moreover, transcriptome-wide association studies (TWAS) analysis revealed the expression of candidate genes associated with meat color traits in specific tissues. Notably, the largest number of candidate genes were observed from transcripts derived from adipose, liver, lung, spleen tissues, and macrophage cell type, indicating their crucial role in meat color development. Several shared genes associated with redness, yellowness, and chroma indices traits were identified, including *RINL* in adipose tissue, *ENSSSCG00000034844* and *ITIH1* in liver tissue, *TPX2* and *MFAP2* in lung tissue, and *ZBTB17*, *FAM131C*, *KIFC3*, *NTPCR*, and *ENGSSSCG00000045605* in spleen tissue. Furthermore, single-cell enrichment analysis revealed a significant association between the immune system and meat color. This finding underscores the significance of the immune system associated with meat color. Overall, our study provides a comprehensive analysis of the genetic mechanisms underlying meat color traits, offering valuable insights for future breeding efforts aimed at improving meat quality.

## 1. Introduction

The color of meat is of paramount importance in evaluating meat quality and shaping consumer preferences serving as a vital indicator for meat evaluation. In addition, meat color provides valuable information regarding meat quality, freshness, nutritional properties, and storage conditions. Consumers usually rely on their understanding of meat color and quality to make informed purchasing decisions [1]. The color of meat can be objectively described using the parameters in the Commission Internationale de l’Éclairage lab (CIELAB) color space. These parameters contain the brightness (L*), redness (a*), yellowness (b*), chroma indices value (C*), and hue angle (h°) of meat color. The CIELAB model is highly intuitive for color evaluations, aligning with the natural human perception of colors. In the meanwhile, it has the advantages of time-saving, preciseness, and reproducibility [2].

Earlier studies (e.g., Esfandyari et al. [3], Lee et al. [4], and Zha et al. [5]) have suggested that meat color traits in pigs demonstrate low to moderate heritability, indicating the significant contribution of genetic factors to the variation in these traits. Therefore, through a comprehensive analysis of the genetic mechanisms underlying meat color traits and the identification of relevant candidate sites and genes, which can in turn enhance the efficiency of marker-assisted selection or genomic selection, valuable insights for the genetic improvement of the traits can be provided [6,7,8,9]. According to the release 48 version of the animal quantitative trait loci (QTLs) database [10], a total of 18,265 QTLs or associations for porcine meat quality and carcass traits have been reported in previous studies. However, only 904 QTLs or associations specifically related to meat color traits have been identified. Meat color traits are complex quantitative traits influenced by multiple genes, and further research is required to unveil their intricate genetic basis. Deeper investigations are essential to gain a comprehensive understanding of the underlying genetic mechanisms governing meat color traits.

In recent years, the application of genetic methods in the study of meat color traits has made notable advancements, with genome-wide association studies (GWAS) being particularly significant. GWAS has emerged as a powerful tool in the last decade, aiming to identify associations between genotypes and phenotypes of various traits or diseases [11,12,13,14,15]. Several GWAS analyses have been conducted on meat color in pigs [16,17,18,19,20]. However, these studies are basically limited in the marker density, resulting in the loss of crucial information. Furthermore, these studies lack comprehensive post-GWAS analyses, e.g., transcriptome-wide association studies (TWAS) and single-cell enrichment analysis, which restrict the depth of research findings. TWAS is a method that integrates transcriptome data and GWAS data to analyze the genetic mechanisms of traits at the transcriptome level. TWAS aims to identify associations between gene expression and a phenotype and is frequently employed as a secondary analysis of GWAS results [21]. Single-cell RNA sequencing (scRNA-seq) has emerged as the preferred method for quantifying gene expression at the single-cell level, enabling researchers to gain a more precise understanding of gene activity across different cell types and offers a higher resolution compared to studies that utilize bulk transcriptomic data [22]. Moreover, scRNA-seq has the ability to identify novel cell populations within traditionally defined cell types [23]. By integrating scRNA-seq with GWAS data, researchers can potentially uncover the crucial tissues, cell types, and cell populations through which candidate variants influence traits or diseases [24,25,26]. The use of TWAS and single-cell transcriptomics for the genetic analysis of meat color traits will be helpful to further understand the genetic structure and regulatory mechanism of meat color.

This study presents a multi-layered analysis to explore the genetic mechanisms underlying meat color traits in pigs (Figure 1). Initially, we estimated the genetic parameters and performed whole-genome sequence-based GWAS on five meat color traits, i.e., L*, a*, b*, C*, and h°. This process enabled the identification of candidate SNPs and genes associated with meat color. Subsequently, we explored selection signatures in pig breeds with distinct meat color characteristics to identify SNPs linked to color variation. Moreover, we conducted TWAS and single-cell enrichment analysis to identify genes and cell types significantly associated with meat color.

## 2. Results and Discussion

### 2.1. Estimates of Heritability and Genetic Correlations

The average values of L*, a*, b*, C*, and h° were 47.78, 7.48, 9.70, 9.85, and 0.90 in Jinhua × Piétrain F_2_ pigs; meanwhile, the values were 40.48, 4.65, 5.67, 7.46, and 0.92 in (Piétrain × Duroc) × (Landrace × Yorkshire) pigs. This study found that meat color traits exhibit low-to-moderate heritability. The heritabilities of L*, a*, b*, C*, and h° are 0.17 (±0.09), 0.14 (±0.10), 0.17 (±0.10), 0.10 (±0.09), and 0.30 (±0.10). This is consistent with the results of previous reports that the heritability of meat color traits is in the range from 0.10 to 0.50, and meat color traits are characterized by low to moderate heritability [3,5]. Meanwhile, we found most of the traits had lower estimates of heritability than in purebred pigs, e.g., Duroc [27] and Berkshire [4]. The previous studies reported varying estimates of heritability between crossbred and purebred data, but generally, heritability estimates tend to be higher for purebred pigs compared to crossbred pigs [3,28,29]. The phenotypic and genetic correlations (Table 1) between a* and h° exhibited a significant negative association. In addition, the phenotypic and genetic correlations between a* and C* revealed a significant positive association, and b* and C* also displayed a significant positive correlation. It is worth noting that the phenotypic and genetic correlations between the traits a* and C*, a* and h°, b* and C*, and b* and h° are generally strong. This could be attributed to the fact that the values of C* and h° are generated by the ratio of the values of a* and b* in the analysis.

### 2.2. GWAS and Selection Signatures for Meat Color Traits

Subsequently, a comprehensive analysis was undertaken to identify promising SNPs and candidate genes associated with meat quality traits. Our findings revealed significant associations between specific SNPs and five meat color traits, i.e., L*, a*, b*, C*, and h° (Figure 2A and Appendix A). These SNPs were found to overlap with a set of candidate genes, with 14, 37, 75, and 114 genes associated with each respective meat color trait (Appendix A). Notably, our QTL enrichment analysis highlighted a strong enrichment of meat and carcass-related QTLs among the SNPs associated with meat color traits (Appendix A). These QTLs encompassed key attributes, e.g., meat color, tenderness score, and fat androstenone levels. These results provide evidence of a relationship between the candidate SNPs and meat color. Additionally, our analysis revealed a significant enrichment of QTLs related to lipid deposition, as lipid oxidation is a crucial factor influencing meat color [2,30]. Interestingly, we also observed the significant enrichment of health-related QTLs that displayed a close relationship with meat color, such as aspartate aminotransferase activity and eosinophil number. Aspartate aminotransferase, also known as glutamic oxaloacetic transaminase, is an enzyme with widespread presence and multiple functions. Its biological functions can influence the changes in glycolytic and apoptotic potentials that occur during the postmortem period. These potentials are crucial in determining the variation in meat quality attributes, e.g., the development of pale-like characteristics [31,32]. Furthermore, the notable enrichment of QTL related to the eosinophil number suggests a connection between immunity and meat color traits. Previous studies have established associations between immunity and production traits [33,34]. These associations present a challenge for breeding programs and necessitate careful consideration in order to enhance immunocompetence without compromising production traits in pigs. Furthermore, we identified a total of 6, 9, 6, 8, and 11 lead SNPs (Table 2) associated with L*, a*, b*, C*, and h°, respectively. These lead SNPs exhibited the lowest *p*-values within their respective genomic regions and are particularly suitable for enhancing the selection models by incorporating prior knowledge. Their inclusion might enhance the efficacy of selecting desirable meat color traits in pigs.

In the current study, we discovered a group of 32 shared SNPs that were associated with at least two meat color traits (Figure 2B). These SNPs were found on chromosomes 6, 10, and 14, with the majority being located on chromosome 6. By utilizing these SNPs, we observed that the five traits could be clustered into two distinct groups: one comprising L* and h°, and the other consisting of a*, b*, and C*. Notably, this clustering pattern aligned with the genetic correlations observed among these traits.

To pinpoint the promising SNPs associated with meat color traits, we conducted analysis to detect selection signatures. The previous study has reported that Laiwu pigs displayed significantly different color parameters compared to Erhualian and Bamaxiang pigs [35]. Therefore, in our analysis, we focused on assessing the genetic differentiation between two groups: Group 1 (Laiwu vs. Erhualian) and Group 2 (Laiwu vs. Bamaxiang). We identified a total of 624 candidate signatures, corresponding to 182 putative genes in Group 1 (Appendix A), and 1094 candidate signatures, corresponding to 307 putative genes in Group 2 (Appendix A). To gain insights into the biological functions and pathways of these putative genes, we further performed gene enrichment analysis (Table 3). Although the identification of selection signatures was broad-spectrum, some of these GO terms and KEGG pathways were related to the formation of meat color, e.g., porphyrin metabolism. Iron will fall off from the porphyrin ring in the heme destruction process, which can further cause a more serious lipid oxidation in muscle foods than heme iron [36], and influence the formation of meat color.

Through the integration of GWAS and selection signature detection, we identified two shared genes, *GIN1* and *PPIP5K2*, which are the newly reported genes associated with meat color traits across different pig breeds. Moreover, we discovered a promising SNP (6_69103754) located within the *GIN1*-*PPIP5K2* genomic region, which exhibited the highest values of *F*_ST_ and θ_π_ (Figure 2C). To further investigate, we examined the allele frequency of 6_69103754 in various pig populations. Our findings revealed a significantly higher frequency of the T allele in both LWU and JS pig populations compared to the commercial pig breeds (Appendix A). These allele frequency results align with the observed phenomenon of LWU and JS pigs exhibiting different and more favorable meat color in comparison to PI, YY, LL, and DD pigs.

### 2.3. TWAS for Meat Color Traits

Furthermore, we investigated the genetic mechanisms involved in meat color formation at the transcriptome level. To initiate this analysis, we conducted a summary-based TWAS in 34 tissues using the FUSION TWAS pipeline. This approach prioritized 14, 35, 45, 47, and 70 candidate genes for L*, a*, b*, C*, and h° traits (Appendix A), respectively. Notably, we observed that candidate genes in various tissues, as the most candidate genes detected in adipose, liver, lung, spleen tissues, and macrophage, were closely related to meat color traits (Figure 3). This suggests that these specific tissues and cell types play a significant role in the development of meat color. In this study, we specifically focused on the significant genes identified in adipose, liver, lung, and spleen tissues.

We discovered that the expression levels of 2, 5, 10, 11, and 19 genes in adipose, liver, lung, and spleen tissues were significantly associated with L*, a*, b*, C*, and h° traits. Notably, the genes *ATXN10* in adipose tissue and *TPX2* in lung tissue reached the significance threshold for L*. Additionally, we identified several shared genes that were related to a*, b*, and C* traits. These include *RINL* in adipose tissue, *ENSSSCG00000034844* and *ITIH1* in liver tissue, *TPX2* and *MFAP2* in lung tissue, and *ZBTB17*, *FAM131C*, *KIFC3*, *NTPCR*, and *ENGSSSCG00000045605* in spleen tissue. Among these traits, we observed that a higher number of genes were significantly associated with the h° trait, which is consistent with the results of GWAS. Notably, *ENSSSCG00000051369* in liver tissue was a shared gene significantly related to both h° and a*. Of all the significant genes associated with h°, the expression level of *RGS14* in liver tissue potentially influences fat deposition, which in turn contributes to the differences in meat color. *RGS14* has been found to play crucial regulatory roles in liver damage and inflammatory responses [37]. In vivo and in vitro studies have demonstrated that overexpression of *RGS14* can effectively affect lipid accumulation, inflammatory response, and liver fibrosis in hepatocytes [38]. Furthermore, we revealed the potential biological functions of *ENSSSCG00000034844*, *ENGSSSCG00000045605*, and *ENSSSCG00000051369* in the formation of meat color in pigs.

Compared to other tissues, muscle tissue exhibits a relatively lower number of candidate genes. There are two potential reasons for this observation. Firstly, it is possible that meat color as measured on the CIELAB scale primarily relies on other tissues, such as fat. Secondly, the lack of subdivision of muscle tissue in the current PigGTEx portal may affect the efficacy of TWAS focusing on muscle tissue. In future investigations, the subdivision of muscle tissue could be considered to elucidate the specific underlying factors.

### 2.4. Single-Cell Enrichment for Meat Color Traits

To identify the potential contributing cell types, we conducted further analysis by utilizing scRNA-seq data from adipose (Appendix A), liver (Appendix A), lung (Appendix A), and spleen (Appendix A) tissues for single-cell enrichment analysis. We integrated the GWAS data for five meat color traits with the single-cell RNA data from these tissues. We present the enrichment results at the cell-type level, aggregated for each cell type based on the individual cell-level results. The findings for a representative subset of cell types from the four tissues and five traits are shown in Figure 4. Within this subset, the scDRS method identified nine cell-type trait associations (FDR < 0.1) and revealed significant heterogeneity in trait associations within specific cell types for two out of the nine identified cell-type trait associations. 

Regarding the associations between cell types and traits, we observed that scDRS analysis consistently linked metabolic and immune cell types with meat color traits. Notably, erythroid cells and hepatocytes in the liver, macrophages in the lung and spleen, plasma cells in the lung, and monocytes and NK cells in the spleen showed significant associations. This suggests the involvement of immune response and metabolic processes in the formation of meat color. Furthermore, the significant correlation between macrophages and meat color highlights the crucial role of the immune response in this process. It is worth noting that the oxidative process in muscle tissues can vary depending on an animal’s immune status [39], as lipid oxidation can impact meat color [2,30]. The results obtained from both single-cell enrichment analysis and QTL enrichment analysis were found to be consistent, indicating an association between meat color and immune system.

Meat color is influenced by multiple factors, including genetics, nutrition, management practices, environmental conditions, and slaughter age [18,19]. For example, the meat color of young animals tends to be paler in comparison to that of adult animals [40]. Genetics has been identified as a particularly influential factor among these variables [18]. Consequently, understanding the genetic mechanisms underlying meat color traits holds significant potential for enhancing meat color.

## 3. Materials and Methods

### 3.1. Samples and Data

All experimental procedures were carried out in accordance with the guidelines of the China Council on Animal Care. The protocol was approved by the Animal Care and Use Committee of Zhejiang University (permit number: ZJU20160346).

This study used the genotypes and phenotypes of a Jinhua × Piétrain F_2_ population generated as follows. Six Piétrain (three boars and three sows) and five Jinhua (three sows and two boars), constituting the F_0_ generation, were successfully mated and produced the F_1_ generation (i.e., six boars and twenty-three sows). The individuals of the F_1_ generation were intercrossed to produce the F_2_ generation. The pigs were provided with standardized care and given unrestricted access to food and water. A total of 288 Jinhua × Piétrain F_2_ pigs (Population 1, P1) were slaughtered with a mean age of 215.4 ± 31.2 days and an average weight of 81.1 ± 11.5 kg in commercial slaughterhouses. The *longissimus dorsi* muscle at the thoracolumbar junction was immediately isolated post-slaughter and stored at −20 °C for subsequent DNA extraction. In addition, we collected a publicly available dataset (Population 2, P2) containing 669 (Piétrain × Duroc) × (Landrace × Yorkshire) pigs [5] to increase the sample size and diversity, thereby enhancing the power of genetic analysis.

The color of *Longissimus lumborum* of 288 Jinhua × Piétrain F_2_ pigs was measured using a bench spectrophotometer (SP60, X-Rite, Grand Rapids, MI, USA) over black and white backgrounds (ColorChecker; X-Rite) in the CIELAB color space with a measure area diameter of 8 mm and 10° observer angle at 45 min of blooming. The lightness (L*), redness (a*), and yellowness (b*) were recorded from the spectrophotometer, as hue angle (h°) and chroma indices (C*) were calculated according to the following formula: h° = tan^−1^ (b*/a*), and C* = (a*^2^ + b*^2^)^0.5^ [41].

All Jinhua × Piétrain F_2_ pigs were genotyped using GeneSeek GGP-Porcine 80 k SNP BeadChip (Neogen Corporation, Lansing, MI, USA). Quality control was performed using PLINK (v1.9) software [42], following the below criteria: (1) retaining the SNPs located in autosomes; (2) removing the SNPs with a call rate less than 90% or minor allele frequency (MAF) less than 0.01. After quality control, the P1 had a total of 240 pigs with full phenotypic records and 49,432 informative SNPs. Meanwhile, the P2 had 669 (Piétrain × Duroc) × (Landrace × Yorkshire) pigs with the meat color parameters (i.e., L*, a*, b*, C*, and h°) and 16,943,752 SNPs. To obtain genotype data at the whole-genome sequence level, we performed genotype imputation for two datasets using the multi-breed Pig Genomics Reference Panel (PGRP v1) from PigGTEx [43]. Initially, we utilized conform-gt program to revise strand inconsistencies of SNPs. Subsequently, we employed the BEAGLE (v5.1) software [44,45] to impute the genotype data of three populations to the sequence level. We filtered out the SNPs with dosage R-squared values less than 0.8. Finally, SNPs with MAF lower than 0.01 were removed, resulting in a total of 909 individuals with 22,914,705 SNPs retained for subsequent genetic analysis.

### 3.2. Statistical Analysis

We fitted a linear mixed model to estimate the heritability and genetic correlation using the average information algorithm [46], as follows:Y=Xβ+Za+Wk+e
where *Y* is the vector of phenotypic observations for L*, a*, b*, C*, and h° obtained from the color measurements; β is the vector of fixed effects, including sex (two levels) and project-batch (ten levels); k is the vector of covariates, including carcass weight at slaughter and top five principal components; *a* is the vector of direct additive effects and is set as a~*N* (0, *G*$\sigma_{g}^{2}$), where *G* is the genomic relationship matrix; e is the vector of residual random effects and is set as e~*N* (0, *I*$\sigma_{e}^{2}$); *I* is an identity matrix; X, Z, and W are incidence matrices for β, a, and k.

The heritability was calculated as follows:$$h^{2}=\frac{\sigma_{g}^{2}}{\sigma_{g}^{2}+\sigma_{e}^{2}}$$
where $h^{2}$ is heritability, $\sigma_{g}^{2}$ is the additive genetic variance, and $\sigma_{e}^{2}$ is the residual variance.

Genetic correlations between meat color traits were calculated as follows:$$r_{g}=\frac{Cov\left(T1,T2\right)}{\sqrt{\left(\sigma_{T1}^{2}\times \sigma_{T2}^{2}\right)}}$$
where $r_{g}$ represents the genetic correlation, $\sigma_{T1}^{2}$,$\;\sigma_{T2}^{2}$, and $Cov\left(T1,T2\right)$ indicate genetic variance and covariance between the breeding values of two traits.

We used a single-marker regression mixed linear model to investigate the association between each SNP and the phenotype of each trait by GEMMA (v0.98) software [47] as follows:y=Wα+Qk+Ub+Su+e
where y is the vector of L*, a*, b*, h°, and C* of each pig; *α* is the vector of the fixed effects, including sex (two levels) and project-batch (ten levels); k is the vector of covariates, including carcass weight at slaughter and top five principal components; b is the vector of the substitution effect of the SNPs; u is the vector of random additive genetic effects and is set as u~*N* (0, *G*$\sigma_{u}^{2}$), where *G* is the genomic relationship matrix [48]; W, Q, U, and S are incidence matrices for b, α, and u; e is the random residuals and is set as e~*N* (0, *I*$\sigma_{e}^{2}$), where *I* is an identity matrix. Furthermore, SNPs with a *p*-value lower than 1/N were considered as genome-wide putative candidate genes affecting meat color traits.

### 3.3. SNP Annotations

QTL enrichment analyses to annotate the putative candidate SNPs based on the Animal QTL Database (Release 48) [10] was performed using GALLO (v1.3) package [49]. Here, QTLs with an FDR less than 0.05 were retained. We designated the SNP with the lowest *p*-value on each chromosome as the lead SNP, and the SNP with the smallest *p*-value outside a 0.4 Mb region upstream and downstream of the primary lead SNP as the secondary lead SNP. This process was repeated until no significant SNPs remained on that chromosome. We extracted the putative candidate SNPs from the Pig HAplotype Reference Panel (v3) server [50] to investigate the differentiation of allele frequency between different pig breeds via the detection of selection signatures, i.e., *F*_ST_ [51] and θ_π_. We conducted the detection of selection signatures in two groups (Group1: Laiwu vs. Erhualian; Group2: Laiwu vs. Bamaxiang). Furthermore, we analyzed the allele frequency of the promising candidate SNP in various pig breeds, i.e., the indigenous Laiwu pig (LWU), the crossbred Jishen black pig (JS), and four commercial pig breeds (Piétrain, PI; Yorkshire, YY; Landrace, LL; Duroc, DD).

The genes located in the genomic regions of selection signatures were identified as candidate selected genes through mapping analysis with the pig reference genome. To further investigate the biological functions of these genes, we conducted GO term and KEGG pathway enrichment analyses of these candidate genes using the DAVID website (v2023q4) with the gene background of *sus scrofa* [52,53].

### 3.4. Transcriptome-Wide Association Study

To explore whether the overall cis-genetic component of the molecular phenotype is associated with meat color traits in 34 tissues, we performed single-tissue TWAS using FUSION method [54] based on GWAS summary statistics on the FarmGTEx TWAS-Server (v1, https://twas.farmgtex.org/, accessed on 1 December 2023) [55]. The GWAS summary statistics files of meat color traits, comprising columns of the chromosome, position, SNP name, effect allele, non-effect allele, *p*-value, and beta coefficient, were uploaded to the server, then the association between the genetically regulated levels of gene expression and the phenotypes with the imputed gene expression level was quantified. Bonferroni correction was used and *p*-value < 0.05 after correction was considered as significance.

### 3.5. Polygenic Signals on Individual Cells

We downloaded the publicly available scRNA-seq datasets of subcutaneous adipose, spleen, lung, and liver [56]. In the datasets, the five regions of the liver (i.e., left lateral lobe, left medial lobe, right medial lobe, right lateral lobe, and quadrate lobe) and seven regions (i.e., left apical lobe, left middle lobe, left main lobe, right apical lobe, right middle lobe, accessory lobe, and right main lobe) of the lung were mixed to generate the scRNA-seq data. Cells were imported into Scanpy (v1.9.5) [57] and filtered based on the following criteria: the number of detected genes were greater than 200 and less than 5000, and the percentage of mitochondrial transcripts from specific mitochondrial genes (*ATP6*, *ATP8*, *COX1*, *COX2*, *COX3*, *CYTB*, *ND2*, *ND3*, *ND4*, *ND4L*, *ND5*, and *ND6*) was less than 30%. Then, cells were clustered using the Leiden algorithm to perform lineage clustering. We selected the top 1000 genes as putative meat color trait-related genes based on gene-level association *p*-values from the TWAS results of the corresponding tissues. We used scDRS (v1.0.2) [26] to quantify the aggregate expression of the putative genes derived from the TWAS results in each cell of scRNA-seq data. Consequently, we generated cell-specific trait-related scores via the function “compute-score”. In brief, 1000 sets of cell-specific raw control scores were calculated from the matched control gene sets. Then, we normalized the raw trait-related score and raw control scores for each cell, producing the normalized trait-related score and normalized control scores. We performed cell-type-level analyses to identify trait-related associations within a predefined cell type using the function “perform-downstream” with default settings. To correct for multiple testing, FDR was calculated via the Benjamini–Hochberg method across all pairs of cell types and five meat color traits.

## 4. Conclusions

In summary, our comprehensive analysis has provided valuable insights into the genetic basis of meat color in pigs. Our investigations led to the identification of SNP 6_69103754 as a promising candidate with varying allele frequencies in pigs with different meat color characteristics. This SNP has the potential to influence the redness and chroma indices values of meat. Furthermore, our TWAS analysis uncovered the expression of several shared candidate genes associated with meat color traits in their respective tissues. Additionally, our single-cell enrichment analysis revealed a significant association between the immune system and meat color. These findings have the potential to advance pig breeding and production, facilitating the development of efficient genomic selection schemes for the genetic improvement of meat color traits.

## Figures and Tables

**Figure 1 ijms-25-03682-f001:**
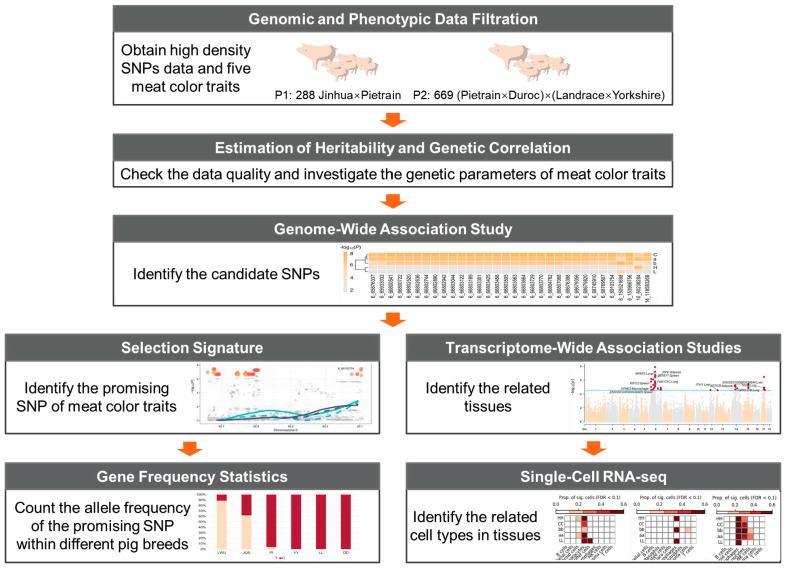
The comprehensive analysis of genetic mechanisms underlying meat color in pigs.

**Figure 2 ijms-25-03682-f002:**
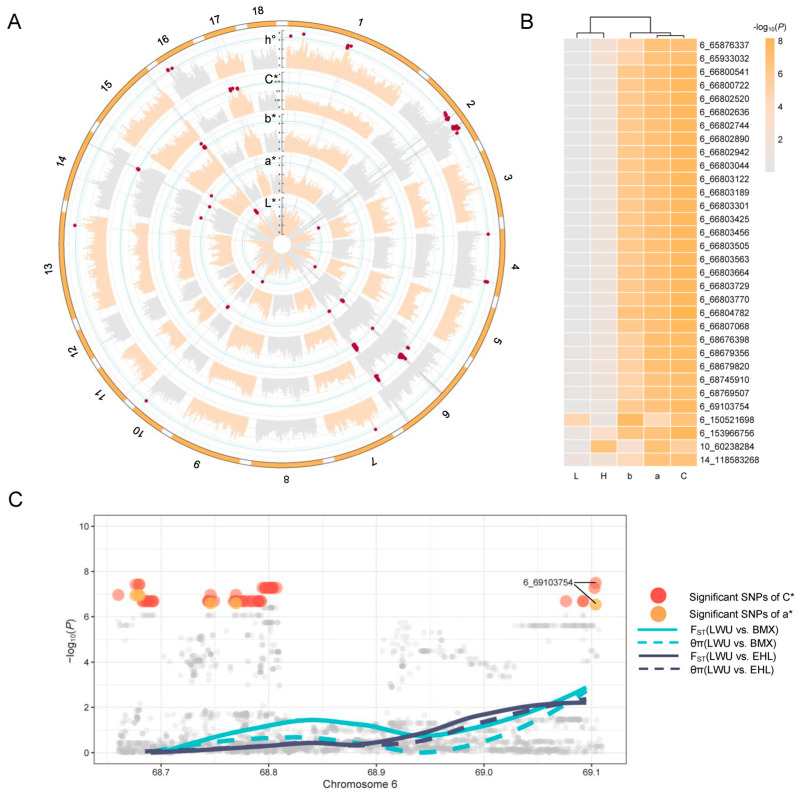
The GWAS and selection signatures of meat color traits. (**A**) The results of GWAS on meat color traits; (**B**) the shared SNPs that were associated with at least two meat color traits; (**C**) the promising candidate SNP located in the GIN1−PPIP5K2 genomic region.

**Figure 3 ijms-25-03682-f003:**
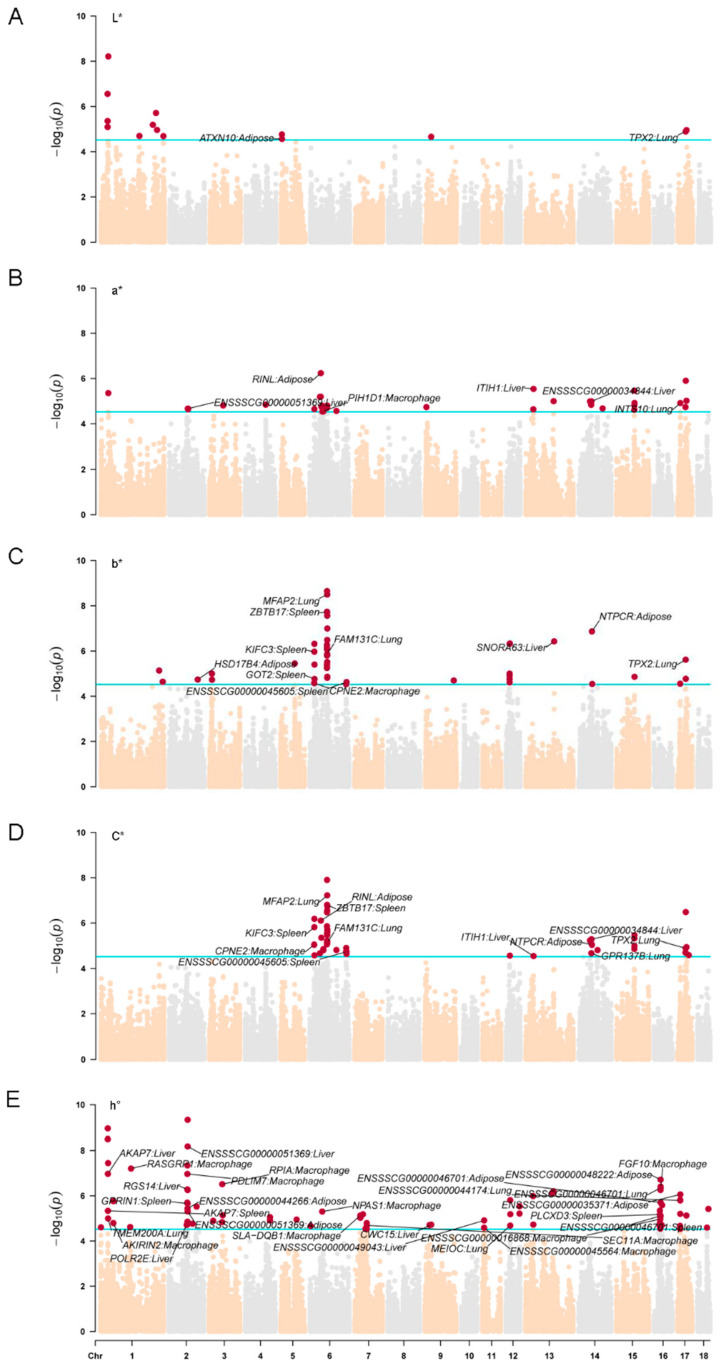
The Manhattan plots of meat color traits: (**A**) L*, (**B**) a*, (**C**) b*, (**D**) C*, and (**E**) h°.

**Figure 4 ijms-25-03682-f004:**
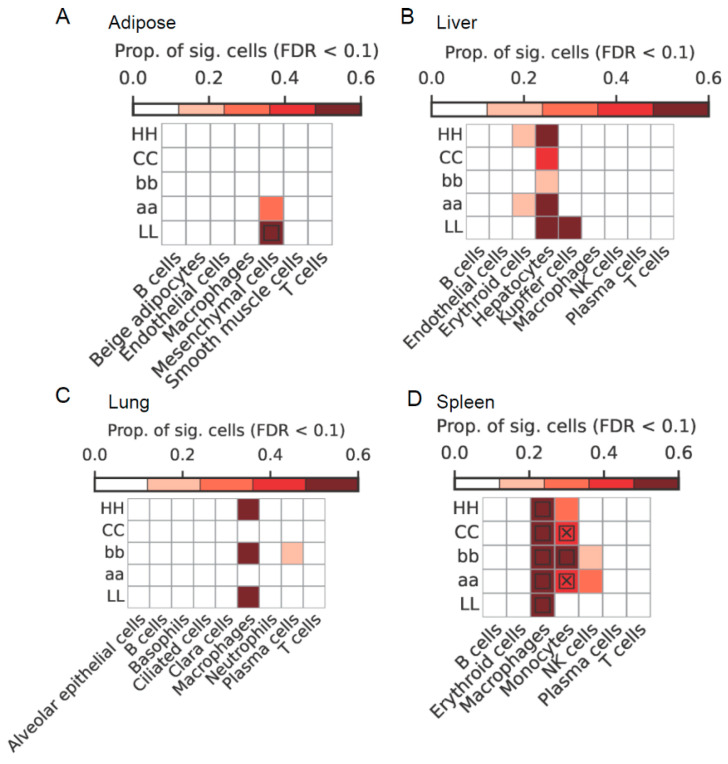
Trait associations at the cell-type level of four tissues: (**A**) adipose, (**B**) liver, (**C**) lung, and (**D**) spleen. The heatmap colors represent the proportion of cells that are significantly associated with each cell-type trait pair; squares indicate significant associations between cell types and traits (FDR < 0.1 across all combinations of cell types and traits); cross symbols indicate significant heterogeneity in the association with traits among individual cells within a specific cell type.

**Table 1 ijms-25-03682-t001:** The phenotypic and genetic correlations between meat color traits.

Correlation	L*	a*	b*	C*	h°
L*		0.04	0.59	0.47	0.31
a*	−0.19 (0.15)		0.75	0.95	−0.81
b*	0.35 (0.41)	0.09 (0.49)		0.91	−0.30
C*	−0.39 (0.55)	0.75 (0.22)	0.73 (0.23)		−0.63
h°	0.58 (0.25)	−0.85 (0.12)	0.42 (0.36)	−0.30 (0.38)	

Note: The diagonals are estimates of genetic and phenotypic correlations between corresponding traits. Below diagonal are genetic correlations, and above diagonal are phenotypic correlations.

**Table 2 ijms-25-03682-t002:** The lead SNPs associated with meat color traits in pigs.

Trait	Chr	SNP	Position	MAF	Beta	*p*-Value
L*	3	3_6002588	6,002,588	0.02	−4.15	2.58 × 10^−7^
	6	6_2992189	2,992,189	0.43	−0.94	2.74 × 10^−7^
	9	9_122912114	122,912,114	0.04	−2.53	9.35 × 10^−8^
	11	11_6188381	6,188,381	0.01	4.17	2.81 × 10^−7^
	15	15_135164457	135,164,457	0.10	−1.96	5.52 × 10^−8^
	15	15_135250924	135,250,924	0.10	−1.86	9.75 × 10^−8^
a*	6	6_65876337	65,876,337	0.08	1.12	1.85 × 10^−7^
	6	6_66804782	66,804,782	0.05	1.57	9.92 × 10^−8^
	6	6_68676398	68,676,398	0.05	1.49	1.14 × 10^−7^
	6	6_69103754	69,103,754	0.05	1.50	2.91 × 10^−7^
	6	6_153966756	153,966,756	0.42	−0.63	2.55 × 10^−7^
	10	10_60238284	60,238,284	0.08	1.07	1.24 × 10^−7^
	14	14_48805995	48,805,995	0.44	−0.67	4.83 × 10^−8^
	14	14_118583268	118,583,268	0.16	0.97	2.92 × 10^−7^
	15	15_6662644	6,662,644	0.08	1.21	3.82 × 10^−8^
b*	6	6_66804782	66,804,782	0.05	0.98	2.67 × 10^−7^
	6	6_150521698	150,521,698	0.15	0.66	3.11 × 10^−8^
	6	6_155031955	155,031,955	0.12	0.70	1.30 × 10^−8^
	6	6_155302633	155,302,633	0.12	0.67	4.40 × 10^−8^
	15	15_130829530	130,829,530	0.14	−0.85	1.74 × 10^−8^
	15	15_134038873	134,038,873	0.11	0.67	1.24 × 10^−7^
C*	6	6_66800541	66,800,541	0.05	1.65	2.21 × 10^−8^
	6	6_153966756	153,966,756	0.42	−0.72	1.48 × 10^−8^
	14	14_118583268	118,583,268	0.16	1.03	2.32 × 10^−7^
	14	14_119428840	119,428,840	0.12	1.15	2.82 × 10^−7^
	14	14_120357132	120,357,132	0.12	1.18	1.38 × 10^−7^
	17	17_16141992	16,141,992	0.02	2.40	8.41 × 10^−8^
	17	17_23084166	23,084,166	0.02	2.43	9.99 × 10^−8^
	17	17_34758105	34,758,105	0.06	−1.46	1.92 × 10^−7^
h°	1	1_6615079	6,615,079	0.05	−0.10	1.39 × 10^−7^
	1	1_32736620	32,736,620	0.39	0.05	3.30 × 10^−8^
	1	1_125247464	125,247,464	0.40	−0.06	7.99 × 10^−8^
	1	1_131796623	131,796,623	0.03	0.13	6.24 × 10^−8^
	2	2_71469230	71,469,230	0.05	0.10	6.97 × 10^−8^
	2	2_83698362	83,698,362	0.47	−0.05	8.05 × 10^−8^
	2	2_106827384	106,827,384	0.06	−0.09	3.47 × 10^−8^
	2	2_112653547	112,653,547	0.09	−0.09	2.35 × 10^−8^
	4	4_9073628	9,073,628	0.02	−0.14	3.00 × 10^−7^
	4	4_106967798	106,967,798	0.11	−0.08	8.98 × 10^−8^
	7	7_24660161	24,660,161	0.22	−0.05	2.22 × 10^−7^

**Table 3 ijms-25-03682-t003:** The enrichment analysis of the selected genes between pigs with distinct meat color.

Group	Term/Pathway	FDR
Group 1	Bone mineralization (GO:0030282)	2.21 × 10^−4^
	Signal transduction (GO:0007165)	2.71 × 10^−4^
	Protein localization (GO:0008104)	3.12 × 10^−4^
	Regulation of exocytosis (GO:0017157)	3.70 × 10^−3^
	Cuticle development (GO:0042335)	2.50 × 10^−3^
	Pentose and glucuronate interconversions (ssc00040)	9.33 × 10^−6^
	Steroid hormone biosynthesis (ssc00140)	8.12 × 10^−5^
	Thyroid hormone synthesis (ssc04918)	9.33 × 10^−5^
	Ascorbate and aldarate metabolism (ssc00053)	2.12 × 10^−3^
Group 2	Anterior neural tube closure (GO:0061713)	7.14 × 10^−4^
	Bicellular tight junction assembly (GO:0070830)	8.12 × 10^−4^
	Regulation of protein stability (GO:0031647)	1.14 × 10^−4^
	Pentose and glucuronate interconversions (ssc00040)	2.91 × 10^−5^
	Bile secretion (ssc04976)	1.12 × 10^−5^
	Ascorbate and aldarate metabolism (ssc00053)	2.23 × 10^−5^
	Steroid hormone biosynthesis (ssc00140)	2.91 × 10^−4^
	Porphyrin metabolism (ssc00860)	7.56 × 10^−3^
	Retinol metabolism (ssc00830)	8.14 × 10^−3^
	Metabolic pathways (ssc01100)	1.23 × 10^−2^
	Biosynthesis of cofactors (ssc01240)	2.37 × 10^−2^

## Data Availability

Data are contained within the article and Appendix A.

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
