# Peer review of "Unveiling the Genetic Mechanism of Meat Color in Pigs through GWAS, Multi-Tissue, and Single-Cell Transcriptome Signatures Exploration"

_ijms, 2024, doi:10.3390/ijms25073682_

Round 1

Reviewer 1 Report

Comments and Suggestions for Authors

The use of so many analysis methods deserves recognition and increases the likelihood of learning about the mechanisms shaping a given feature. However, in order for the obtained results to be understandable to readers, a detailed description of individual methods and the authors' intentions is required. This was missing in this work. I suggest correcting figure 1 (resolution, languge:"identification" instead of "identify" and describe in more detailed ways what was the aim of each method). Also please provide more details on TWAS methodolgy in M&M section. Please provide the details of all downloaded data used in the study (tissue, breed, sex, how many samples)

More detailed comments:

Lines 34-36 Please split the sentence into two.

Lines 39-40 Please indicate in brackets the relevant mark.

Line 42 Add in brackets by whom developed

Line 87 Create a short but precise description of the figure, for example "The comprehensive analysis of genetic factors determining the color of pork meat."

Lines 89-123 I suggest dividing this paragraph into three parts

1.       Samples – provide a detailed description of the number of samples used, briefly describe the conditions of animal maintenance, and the method of sample collection and storage. It's also worth mentioning the average weight of animals before slaughter.

2.       Meat color assessment – specify how many samples were analyzed.

3.       Genotyping.

Line 113 Specify what is meant by "five meat color records."

Line 167 Add the reference like in the rest of the text.

Lines 169-171 Should be placed in the Results and Discussion section.

Lines 176-177 Should be placed in the Results and Discussion section.

Line 182 Please use italics for the Latin name Sus scrofa.

Line 207 Please add the results of the meat colour assessment.

Lines 224-226 Information placed below the table should be included in the table description.

Lines 266-268 Please move to the description of Fig. 2.

Lines 277-289 I suggest, for clarity, to place this information in a table. The table can also highlight processes that may particularly affect meat color.

Line 302 The term "commercial pigs" seems to be an abbreviation of thought. Specify.

Line 303 - Isn't it strange that muscle was not among the tissues related to the color of meat?

Line 306 Expand the abbreviation.

Lines 309-312 Please, try to justify it.

Lines 325-326 "in vitro" and "in vivo" should be written in italics.

Line 330-333 If possible, increase the clarity of the image, especially E

Line 337 Expand the abbreviation.

Lines 358-362 Place the figure description in the appropriate place, similar to lines 266-268.

Line 529 Delete point 58.

Please insert figure 4. before the last paragraph of “Single-cell enrichment for meat colour traits”

Furthermore, I recommend that the authors refer to the already known determinants of meat colour. The authors did not discuss pig diet and housing conditions, slaughter process and storage conditions which appear to be important factors influencing meat

Author Response

Reviewer #1: The use of so many analysis methods deserves recognition and increases the likelihood of learning about the mechanisms shaping a given feature. However, in order for the obtained results to be understandable to readers, a detailed description of individual methods and the authors' intentions is required.

Author: Thank you for your careful work and thoughtful suggestions that have helped improve our paper substantially.

This was missing in this work. I suggest correcting figure 1 (resolution, language: "identification" instead of "identify" and describe in more detailed ways what was the aim of each method).

Author: Thanks for your suggestion. We have modified the Figure 1.

Also please provide more details on TWAS methodology in M&M section.

Author: Thanks for your suggestion. We have supplemented the relevant information. Please see Line 199-203.

“To explore whether the overall cis-genetic component of the molecular phenotype is associated with meat color traits in 34 tissues, we performed single-tissue TWAS using FUSION method [41] based on GWAS summary statistics on the FarmGTEx TWAS-Server (v1, https://twas.farmgtex.org/) [42]. The GWAS summary statistics files of meat color traits comprised columns of the chromosome, position, SNP name, effect allele, non-effect allele, P-value, and beta coefficient were uploaded to the server, then quantified the association between the genetically regulated levels of gene expression and the phenotypes with the imputed gene expression level. Bonferroni correction was used and P-value < 0.05 after correction was considered as significance.”

Please provide the details of all downloaded data used in the study (tissue, breed, sex, how many samples)

Author: Thanks for your suggestion. We have supplemented the relevant description of the downloaded data as detailed as possible.

More detailed comments:

Lines 34-36 Please split the sentence into two.

Author: Thanks for your suggestion. We have split the sentence into two. Please see Line 34-36.

“The color of meat is of paramount importance in evaluating meat quality and shaping consumer preferences Serving as a vital indicator for meat evaluation. In addition, meat color provides valuable information regarding meat quality, freshness, nutritional properties, and storage conditions.”

Lines 39-40 Please indicate in brackets the relevant mark.

Author: Thanks for your suggestion. We have supplemented the relevant mark. Please see Line 38-40.

Line 42 Add in brackets by whom developed

Author: Thanks for your suggestion. We have supplemented the relevant information. Please see Line 43.

Line 87 Create a short but precise description of the figure, for example "The comprehensive analysis of genetic factors determining the color of pork meat."

Author: Thanks for your suggestion. We have revised the description of Figure 1.

“Figure 1. The comprehensive analysis of genetic mechanisms underlying meat color in pigs.”

Lines 89-123 I suggest dividing this paragraph into three parts

  1. Samples – provide a detailed description of the number of samples used, briefly describe the conditions of animal maintenance, and the method of sample collection and storage. It's also worth mentioning the average weight of animals before slaughter.
  2. Meat color assessment – specify how many samples were analyzed.
  3. Genotyping.

Author: Thanks for your suggestion. We divided the section 2.1 into three parts according to your suggestion. Please see Line 93-134.

“All experimental procedures were carried out in accordance with the guidelines of the China Council on Animal Care. The protocol was approved by the Animal Care and Use Committee of Zhejiang University (permit number: ZJU20160346).

This study used the genotypes and phenotypes of a Jinhua × Piétrain F2 population generated as follows. Six Piétrain (three boars and three sows) and five Jinhua (three sows and two boars), constituting the F0 generation, were successfully mated and produced the F1 generation (i.e., six boars and 23 sows). The individuals of the F1 gen-eration were intercrossed to produce the F2 generation. The pigs were provided with standardized care and given unrestricted access to food and water. A total of 288 Jinhua × Piétrain F2 pigs (Population 1, P1) were slaughtered with a mean age of 215.4 ± 31.2 days and the average weight of 81.1 ± 11.5 kg in commercial slaughterhouses. The longissimus dorsi muscle at the thoracolumbar junction was immediately isolated post-slaughter and stored at -20℃ for subsequent DNA extraction. In addition, we collected a publicly available dataset (Population 2, P2) contained 669 (Piétrain × Duroc) × (Landrace × Yorkshire) pigs [5] to increase sample size and diversity, thereby enhancing the power of genetic analysis.

The color of Longissimus lumborum of 288 Jinhua × Piétrain F2 pigs were measured using a bench spectrophotometer (SP60, X-Rite, Grand Rapids, MI, USA) over black and white backgrounds (ColorChecker; X-Rite) in the CIELAB color space with a measure area diameter of 8 mm and 10° observer angle at 45 min of blooming. The lightness (L*), redness (a*), and yellowness (b*) were recorded from the spectrophotometer, as hue angle (h°) and chroma indices (C*) were calculated according to the following formula: h° = tan−1 (b*/a*), and C* = (a*2 + b*2)0.5 [27].

All Jinhua × Piétrain F2 pigs were genotyped using GeneSeek GGP-Porcine 80k SNP BeadChip (Neogen Corporation, Lansing, MI, USA). Quality control was per-formed using PLINK (v1.9) software [28], following the below criteria: (1) retaining the SNPs located in autosomes; (2) removing the SNPs with a call rate less than 90% or minor allele frequency (MAF) less than 0.01. After quality control, the P1 has a total of 240 pigs with full phenotypic records (Table 1) and 49,432 informative SNPs. Mean-while, the P2 has 669 (Piétrain × Duroc) × (Landrace × Yorkshire) pigs with the meat color parameters (i.e., L*, a*, b*, C*, and h°) and 16,943,752 SNPs. To obtain genotype data at the whole-genome sequence level, we performed genotype imputation for two datasets using the multi-breed Pig Genomics Reference Panel (PGRP v1) from PigGTEx [29]. Initially, we utilized conform-gt program to revise strand inconsistencies of SNPs. Subsequently, we employed the BEAGLE (v5.1) software [30,31] to impute the genotype data of three populations to the sequence level. We filtered out the SNPs with dosage R-squared values less than 0.8. Finally, SNPs with MAF lower than 0.01 were removed, resulting in a total of 909 individuals with 22,914,705 SNPs retained for subsequent genetic analysis.”

Line 113 Specify what is meant by "five meat color records."

Author: Thanks for your suggestion. We have revised the relevant information. Please see Line 124.

“… with the meat color parameters (i.e., L*, a*, b*, C*, and h°) and …”

Line 167 Add the reference like in the rest of the text.

Author: Thanks for your suggestion. Revised.

Lines 169-171 Should be placed in the Results and Discussion section.

Author: Thanks for your suggestion. We have moved the sentence to Results and Discussion section. Please see Line 296-298.

Lines 176-177 Should be placed in the Results and Discussion section.

Author: Thanks for your suggestion. We have moved the sentence to Results and Discussion section. Please see Line 329-331.

Line 182 Please use italics for the Latin name Sus scrofa.

Author: Thanks for your suggestion. Revised.

Line 207 Please add the results of the meat colour assessment.

Author: Thanks for your suggestion. We have supplemented the results of the meat color assessment. Please see Line 229-231.

“The average values of L*, a*, b*, C* and h° were 47.78, 7.48, 9.70, 9.85, and 0.90 in Jinhua × Piétrain F2 pigs, meanwhile, the values were 40.48, 4.65, 5.67, 7.46, and 0.92 in (Piétrain × Duroc) × (Landrace × Yorkshire) pigs (Table 1).”

Lines 224-226 Information placed below the table should be included in the table description.

Author: Thanks for your suggestion. Revised.

Lines 266-268 Please move to the description of Fig. 2.

Author: Thanks for your suggestion. Revised.

Lines 277-289 I suggest, for clarity, to place this information in a table. The table can also highlight processes that may particularly affect meat color.

Author: Thanks for your suggestion. Added. Please see Table 4.

Line 302 The term "commercial pigs" seems to be an abbreviation of thought. Specify.

Author: Thanks for your suggestion. Revised. Please see Line 331.

“These allele frequency results align with the observed phenomenon of LWU and JS pigs exhibiting different meat color in comparison to PI, YY, LL, and DD pigs.”

Line 303 - Isn't it strange that muscle was not among the tissues related to the color of meat?

Author: Thanks. The candidate genes detected by TWAS were shown in Table S6. We found that the expression levels of some candidate genes (e.g., ENSSSCG00000044836, TM9SF4, and DNAJC16) in muscle were significantly associated with meat color traits. However, compared to other tissues, the candidate genes are relatively low. Therefore, this study did not emphasize candidate genes in the muscle. Nevertheless, this intriguing observation has been addressed in the revised manuscript with additional discussion. Please see Table S6 and Line 360-365.

“Compared to other tissues, the muscle tissue exhibits a relatively lower number of candidate genes. There are two potential reasons for this observation. Firstly, it is pos-sible that meat color primarily relies on other tissues, such as fat. Secondly, the lack of subdivision of muscle tissue in the current PigGTEx portal may affect the efficacy of TWAS focusing on muscle tissue. In future investigations, the subdivision of muscle tissue could be considered to elucidate the specific underlying factors.”

Line 306 Expand the abbreviation.

Author: Thanks for your suggestion. Revised.

Lines 309-312 Please, try to justify it.

Author: Thanks. We identified the candidate genes significantly associated with meat color traits at the transcriptome level by TWAS in 34 tissues. We observed the presence of candidate genes in various tissues (Table S6), with the highest number of candidates being identified in adipose, liver, lung, spleen tissues, and macrophage.

Lines 325-326 "in vitro" and "in vivo" should be written in italics.

Author: Thanks for your suggestion. Revised.

Line 330-333 If possible, increase the clarity of the image, especially E

Author: Thanks for your suggestion. We have improved the resolution of the figures to ensure a clearer depiction of the results. Please see Figure 3.

Line 337 Expand the abbreviation.

Author: Thanks for your suggestion. Revised.

Lines 358-362 Place the figure description in the appropriate place, similar to lines 266-268.

Author: Thanks for your suggestion. Revised.

Line 529 Delete point 58.

Author: Thanks for your suggestion. Revised.

Please insert figure 4. before the last paragraph of “Single-cell enrichment for meat colour traits”

Author: Thanks for your suggestion. Revised.

Furthermore, I recommend that the authors refer to the already known determinants of meat colour. The authors did not discuss pig diet and housing conditions, slaughter process and storage conditions which appear to be important factors influencing meat

Author: Thanks for your suggestion. We have supplemented the relevant discussion. Please see Line 398-403.

“Meat color is influenced by multiple factors, including genetics, nutrition, man-agement practices, environmental conditions, and slaughter age [18,19]. For example, the meat color of young animals tends to be paler in comparison to that of adult animals [58]. Genetics has been identified as a particularly influential factor among these variables [18]. Consequently, understanding the genetic mechanisms underlying meat color traits holds significant potential for enhancing meat color performance.”

Reviewer 2 Report

Comments and Suggestions for Authors

Unveiling the Genetic Mechanism of Meat Color in Pigs through GWAS, Multi-Tissue, and Single-Cell Transcriptome Signatures Exploration. Int. J. Mol. Sci.

The manuscript is very interesting and has not only scientific value, but may also be used in the future in the practice of pig breeding and selection. Authors used the TWAS and single-cell transcriptomics for genetic analysis of meat color traits (L*, a*, b*, C*, h°) to identify  promising SNPs, candidate genes and cell types significantly associated with meat color, what is helpful to understand the genetic structure and regulatory mechanism of meat color. The results obtained have provided valuable insights into the genetic basis of meat color in pigs. Manuscript needs the minor revision.

Details:

- the inscriptions on the figures are illegible.

- the age of the pigs on the day of slaughter was not similar (215.4 ± 31.2 days) – what was the body weight of the pigs on the day of slaughter?

- it is not clear how many animals were used in the experiment - please provide the number of live animals. Was it only F2 generation?

- how long after slaughter were samples taken?

- where did the adipose tissue samples come from?

- from which lobe of the liver and lung were the samples taken?

- how were the tissues secured immediately after collection and how were they stored until analysis?

- research results indicate that nutrition (or feed additives) as well as the living/production system may affect the color of meat. Were all animals tested in the experiment (F2) fed the same feed mixture and kept in the same conditions? Please, describe both briefly.

Author Response

Reviewer #2: The manuscript is very interesting and has not only scientific value, but may also be used in the future in the practice of pig breeding and selection. Authors used the TWAS and single-cell transcriptomics for genetic analysis of meat color traits (L*, a*, b*, C*, h°) to identify promising SNPs, candidate genes and cell types significantly associated with meat color, what is helpful to understand the genetic structure and regulatory mechanism of meat color. The results obtained have provided valuable insights into the genetic basis of meat color in pigs. Manuscript needs the minor revision.

Author: Thanks for your constructive suggestion to help improve our paper substantially.

Details:

- the inscriptions on the figures are illegible.

Author: Thanks for your suggestion. We have improved the resolution and size of the figures to ensure a clearer depiction of the results.

- the age of the pigs on the day of slaughter was not similar (215.4 ± 31.2 days) – what was the body weight of the pigs on the day of slaughter?

Author: Thanks for your suggestion. Added. Please see Line 103.

“…the average weight of 81.1 ± 11.5 kg…”

- it is not clear how many animals were used in the experiment - please provide the number of live animals. Was it only F2 generation?

Author: Thanks for your suggestion. We have revised the description of the samples. Please see Line 93-134.

“All experimental procedures were carried out in accordance with the guidelines of the China Council on Animal Care. The protocol was approved by the Animal Care and Use Committee of Zhejiang University (permit number: ZJU20160346).

This study used the genotypes and phenotypes of a Jinhua × Piétrain F2 population generated as follows. Six Piétrain (three boars and three sows) and five Jinhua (three sows and two boars), constituting the F0 generation, were successfully mated and produced the F1 generation (i.e., six boars and 23 sows). The individuals of the F1 gen-eration were intercrossed to produce the F2 generation. The pigs were provided with standardized care and given unrestricted access to food and water. A total of 288 Jinhua × Piétrain F2 pigs (Population 1, P1) were slaughtered with a mean age of 215.4 ± 31.2 days and the average weight of 81.1 ± 11.5 kg in commercial slaughterhouses. The longissimus dorsi muscle at the thoracolumbar junction was immediately isolated post-slaughter and stored at -20℃ for subsequent DNA extraction. In addition, we collected a publicly available dataset (Population 2, P2) contained 669 (Piétrain × Duroc) × (Landrace × Yorkshire) pigs [5] to increase sample size and diversity, thereby enhancing the power of genetic analysis.

The color of Longissimus lumborum of 288 Jinhua × Piétrain F2 pigs were measured using a bench spectrophotometer (SP60, X-Rite, Grand Rapids, MI, USA) over black and white backgrounds (ColorChecker; X-Rite) in the CIELAB color space with a measure area diameter of 8 mm and 10° observer angle at 45 min of blooming. The lightness (L*), redness (a*), and yellowness (b*) were recorded from the spectrophotometer, as hue angle (h°) and chroma indices (C*) were calculated according to the following formula: h° = tan−1 (b*/a*), and C* = (a*2 + b*2)0.5 [27].

All Jinhua × Piétrain F2 pigs were genotyped using GeneSeek GGP-Porcine 80k SNP BeadChip (Neogen Corporation, Lansing, MI, USA). Quality control was per-formed using PLINK (v1.9) software [28], following the below criteria: (1) retaining the SNPs located in autosomes; (2) removing the SNPs with a call rate less than 90% or minor allele frequency (MAF) less than 0.01. After quality control, the P1 has a total of 240 pigs with full phenotypic records (Table 1) and 49,432 informative SNPs. Mean-while, the P2 has 669 (Piétrain × Duroc) × (Landrace × Yorkshire) pigs with the meat color parameters (i.e., L*, a*, b*, C*, and h°) and 16,943,752 SNPs. To obtain genotype data at the whole-genome sequence level, we performed genotype imputation for two datasets using the multi-breed Pig Genomics Reference Panel (PGRP v1) from PigGTEx [29]. Initially, we utilized conform-gt program to revise strand inconsistencies of SNPs. Subsequently, we employed the BEAGLE (v5.1) software [30,31] to impute the genotype data of three populations to the sequence level. We filtered out the SNPs with dosage R-squared values less than 0.8. Finally, SNPs with MAF lower than 0.01 were removed, resulting in a total of 909 individuals with 22,914,705 SNPs retained for subsequent genetic analysis.”

- how long after slaughter were samples taken?

Author: Thanks for your suggestion. The longissimus dorsi muscle at the thoracolumbar junction was immediately isolated post-slaughter and stored at -20℃ for subsequent DNA extraction.

- where did the adipose tissue samples come from?

Author: Thanks. We used the single cell transcriptome dataset of subcutaneous adipose tissue in the current study.

- from which lobe of the liver and lung were the samples taken?

Author: Thanks.

Liver: the fresh liver was collected and sampled with five different anatomical regions: left lateral lobe, left medial lobe, right medial lobe, right lateral lobe, and quadrate lobe. For each region, 1 g tubes were punched and washed twice with cold PBS.

Lung: the samples were mixed and carefully dissected into small pieces. Seven different regions: left apical lobe, left middle lobe, left main lobe, right apical lobe, right middle lobe, accessory lobe, and right main lobe were collected from the fresh porcine lung. From each region, a 0.5 g piece was collected and washed twice in cold HBSS. The samples were dissected into small pieces.

We have supplemented the description of the datasets in the revised manuscript. Please see Line 206-210.

- how were the tissues secured immediately after collection and how were they stored until analysis?

Author: Thanks. The fresh tissues were collected, immediately placed on ice, and processed within 30 min. Each tissue was dissociated and digested independently.

- research results indicate that nutrition (or feed additives) as well as the living/production system may affect the color of meat. Were all animals tested in the experiment (F2) fed the same feed mixture and kept in the same conditions? Please, describe both briefly.

Author: Thanks for your suggestion. We have supplemented the relevant information. Please see Line 100-101

“The pigs were provided with standardized care and given unrestricted access to food and water.”

Round 2

Reviewer 1 Report

Comments and Suggestions for Authors

All my questions have been addressed.

Author Response

Thank you for your careful work and thoughtful suggestions that have helped improve our paper substantially.